# Superconducting phase diagram of $H_3S$ under high magnetic fields

Shirin Mozaffari[1], Dan Sun[2], Vasily S. Minkov[3], Alexander P. Drozdov[3], Dmitry Knyazev[3], Jonathan B. Betts[2], Mari Einaga[4], Katsuya Shimizu[4], Mikhail I. Eremets[3], Luis Balicas[1] & Fedor F. Balakirev[2]

The discovery of superconductivity at 260 K in hydrogen-rich compounds like $LaH_{10}$ re-invigorated the quest for room temperature superconductivity. Here, we report the temperature dependence of the upper critical fields $\mu_0 H_{c2}(T)$ of superconducting $H_3S$ under a record-high combination of applied pressures up to 160 GPa and fields up to 65 T. We find that $H_{c2}(T)$ displays a linear dependence on temperature over an extended range as found in multigap or in strongly-coupled superconductors, thus deviating from conventional Werthamer, Helfand, and Hohenberg (WHH) formalism. The best fit of $H_{c2}(T)$ to the WHH formalism yields negligible values for the Maki parameter $\alpha$ and the spin–orbit scattering constant $\lambda_{SO}$. However, $H_{c2}(T)$ is well-described by a model based on strong coupling superconductivity with a coupling constant $\lambda \sim 2$. We conclude that $H_3S$ behaves as a strong-coupled orbital-limited superconductor over the entire range of temperatures and fields used for our measurements.

[1] National High Magnetic Field Laboratory, Florida State University, Tallahassee, FL 32310, USA. [2] Los Alamos National Laboratory, Los Alamos, NM 87545, USA. [3] Max-Planck-Institut fuer Chemie, Hahn-Meitner Weg 1, 55128 Mainz, Germany. [4] KYOKUGEN, Graduate School of Engineering Science, Osaka University, Machikaneyamacho 1-3, Toyonaka, Osaka 560-8531, Japan. Correspondence and requests for materials should be addressed to F.F.B. (email: fedor@lanl.gov)

The ongoing scientific endeavor to stabilize superconductivity at room temperature led to the discovery of superconductivity, with a very high critical temperature $T_c = 203$ K, in sulfur hydride $H_3S$ under high pressures of $p = 155$ GPa[1]. $H_3S$ along with other hydrides[2,3] seems to be the closest compound, so far, to metallic hydrogen, which is predicted to be a high temperature superconductor[4–6]. $H_3S$ forms as the result of the chemical instability of $H_2S$ under high pressures, where $H_2S$ decomposes into elemental sulfur S and $H_3S$[7–10]. The structure of $H_3S$ is believed to be body-centered cubic with Im-3m symmetry above 150 GPa[10], and characterized by H atoms symmetrically situated midway between two body-centered S atoms. $T_c$ shows a sharp increase from 95 K to 203 K in the pressure range of 110–155 GPa, but decreases with further increasing the pressure due to the pressure-induced phase transition from the R-3m to the Im-3m phase of $H_3S$[1,10].

The superconductivity in $H_3S$ is believed to be conventional, or whose pairing is mediated by phonons characterized by high-frequency optical modes due to the motion of hydrogen[1,11–14]. The high $T_c$ arises from both the metallization of the strongly covalent bonds in $H_3S$[15] and the high phonon frequencies displayed by its light elements[16]. Band structure calculations indicate that $H_3S$ is a multiband metal[17] having a large Fermi surface (broad energy dispersive bands) as the result of the hybridization between the H 1$s$ and the 3$p$ orbitals of sulfur[8,9,13,16,18,19]. Band structure calculations also yield small Fermi surface pockets for the high-$T_c$ phase of $H_3S$[17].

Despite the very high $T_c$s reported for $H_3S$, the studies of $H_{c2}(T)$ are limited to a narrow range of temperatures close to $T_c$ due to the inherent experimental difficulties in performing ultrahigh pressure measurements under very high magnetic fields. The behavior of $H_{c2}(T)$ provides valuable information such as an estimation of the Cooper pair coherence length, the strength of the electron–phonon coupling, the role of the spin–orbit coupling, and the dominant mechanism breaking the Copper pairs.

In type-II superconductors, the orbital and the spin-paramagnetic pair-breaking effects are the two main mechanisms depairing electrons under high magnetic fields. The orbital pair-breaking effect explains the suppression of superconductivity via the formation of Abrikosov vortices[20] in the presence of a field. The superconductivity is suppressed when the kinetic energy associated with vortex currents exceeds the condensation energy of the paired electrons. When the magnetic field approaches a critical value $H_{c2}^{orb} = \phi_0/2\pi\xi^2$, referred to as the orbital limiting field, the vortex cores overlap and the system returns to the normal state. Here, $\phi_0$ is the flux quantum and $\xi$ is the coherence length. The spin-paramagnetic effect explains the effect of the magnetic field on the spin of the electron based on the work by Clogston[21] and Chandrasekhar[22]. When the magnetic energy exceeds the superconducting gap, i.e. $1/2\chi_p H_p^2 = 1/2N(E_F)\Delta^2$, superconductivity will be suppressed, where $\chi_p$, $N(E_F)$, and $\Delta$ are the normal state paramagnetic susceptibility, the density of states at the Fermi level, and the superconducting gap, respectively. If we only consider the spin-paramagnetic effect, the zero-temperature Pauli limiting field (Chandrasekhar-Clogston limit) for a weakly coupled superconductor is approximately $\mu_0 H_p(0) = 1.86\ T_c$, where $2\Delta(T = 0) \sim 3.52\ k_B T_c$ (for a conventional Bardeen-Cooper-Schrieffer (BCS) superconductor) and $\chi_p = g\mu_B^2 N(E_F)$ ($g$ is the Landé $g$ factor and $\mu_B$ is the Bohr magneton). $H_p$ can be renormalized by the strength of the electron–phonon coupling or by electronic correlations. In many superconductors $H_{c2}(T)$ is affected by both pair-breaking mechanisms. The Maki[23] parameter $\alpha = \sqrt{2}H_{c2}^{orb}(0)/H_p(0)$ is a measure of the relative strength between the orbital and the paramagnetic pair-breaking

mechanisms for a given type-II superconductor. The WHH theory[24] includes both pair-breaking effects through the Maki parameter and the spin–orbit constant.

Here, we report measurements of the upper critical field in $H_3S$ samples under extremely high pressures and high magnetic fields. The ultrahigh pressures can only be obtained through the use of a diamond anvil cell (DAC), however, measurements on samples contained by a DAC are quite challenging, particularly in pulsed fields due to the narrow magnet bore and the large magnetic-field induced currents which heat the cell. The small diameter of the DAC[1] employed in our study allows us to perform transport measurements up to 65 T without heating the sample significantly. Our pulsed field measurements under $p > 150$ GPa represent a significant pressure increase over a previous pulsed field study under 10 GPa[25]. We find that $H_3S$ is an orbital-limited superconductor over the entire temperature range and likely a multigap or strongly coupled superconductor.

## Results

**Temperature dependence of the resistance.** Figure 1 displays the resistance as a function of the temperature $T$ for $H_3S$ samples pressurized up to 155 GPa and 160 GPa. The onset, i.e., the first deviation from the normal state resistivity, of the superconducting transitions are estimated to be $T_c = 201$ K for the sample under 155 GPa, and $T_c = 174$ K for the other sample under 160 GPa. The drop in $T_c$ for the higher pressure sample is consistent with the results in ref. [1]. The width of these transitions are $\Delta T_c = 5.5$ K and 26 K for the samples under 155 GPa and 160 GPa, respectively, indicating that the latter sample is less homogeneous, or that it is subjected to stronger pressure gradients. The value of $T_c$ for the sample under 155 GPa increases slightly over time, indicating an evolution towards a more homogeneous sample or weaker pressure gradients. Notice that the superconducting transition displayed by this sample shows a step around 195 K indicating that its superconducting state indeed is inhomogeneous. For samples synthesized within the confines of the DAC a certain degree of inhomogeneity is inevitable. The multiple SC transition steps merge into a single one above a

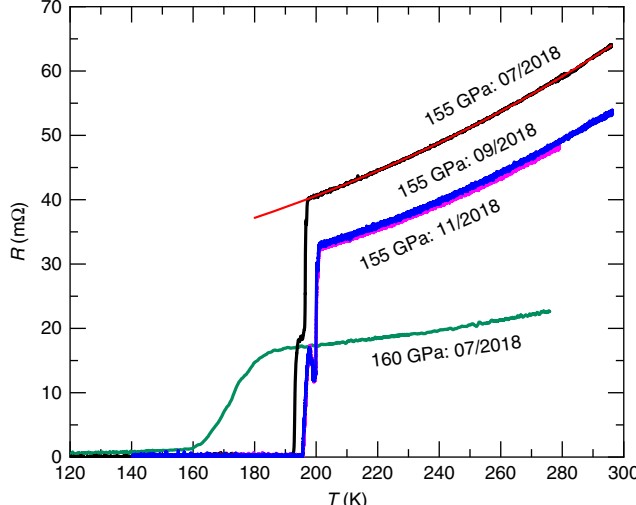

**Fig. 1** Resistive transition towards the superconducting state. Resistance $R$ as a function of the temperature $T$ for two DACs containing $H_3S$ samples under pressures $p = 155$ GPa and 160 GPa. Red line is a fit to $R = R_0 + AT^2$. Notice that the $T_c$ for the 155 GPa sample shifts slightly to higher values while its resistance decreases between July and September 2018. Source data are provided as a Source Data file

moderate field of 3 T in the 155 GPa sample (Supplementary Fig. 1).

We find that $R(T)$ exhibits supralinear behavior in both samples, which is well described by a $R \propto T^2$ leading term. A quadratic temperature dependence for the resistivity is usually ascribed to the inelastic electron–electron scattering in Fermi liquid theory. While our estimates point to a predominantly electron–phonon transport scattering at high temperatures (Supplementary Fig. 2), a linear temperature dependence for the resistivity is typically expected for this regime, hence $R \propto T^2$ is quite unusual. Spectroscopic studies on $H_3S$ attribute the supralinear behavior of $R(T)$ to strong coupling between electrons and a very high energy optical phonon[26]. A similar explanation was given for the supralinear $R(T)$ observed in $MgB_2$, which is another BCS superconductor characterized by a strong coupling to an optical phonon mode[27].

**Hall resistance as a function of the temperature**. Figure 2a displays the Hall resistance $R_{xy}$ for the sample under $p = 155$ GPa as a function of the magnetic field $\mu_0 H$ near the superconducting transition. The $R_{xy}(\mu_0 H)$ do not reveal any non-linearity as a function of $\mu_0 H$ all the way up to 35 T. This indicates that near and above $T_c$ the electrical transport in $H_3S$ is dominated by a single type of carrier (electrons) despite $H_3S$ being predicted to be a multiband metal[17]. Most likely, this indicates that electrons display a considerably higher carrier mobility than holes at these temperatures. The temperature dependence of the Hall coefficient $R_H$ is shown in Fig. 2b, obtained by multiplying the slope of $R_{xy}(\mu_0 H)$ at high fields by the sample thickness, estimated to be $t = (2 \pm 1)$ μm. We observed a noticeable decrease in $R_H(T)$ by as much as 20% as the temperature is decreased, corresponding to an increase in the effective carrier density $n = 1/eR_H$. The smaller values of the Hall coefficient at lower temperatures could come from a decrease in electron mobility, a progressive increase in hole mobility, or from an evolution in their relative densities. If one assumes that only one band contributed to the transport of carriers, the Hall coefficient would yield an electron density $n = 1/eR_H = (8.5 \pm 4.3) \times 10^{22}$ cm$^{-3}$ at room temperature, which is relatively close to known values for transition metals such as copper.

**Superconducting phase diagram**. To determine the temperature dependence of $H_{c2}$, we measured the isothermal resistance as a function of the magnetic field $\mu_0 H$ at selected temperatures ranging from $T = 55$ K to 200 K. Figure 3a, b show the magnetic field dependence of the resistance for the samples under 155 GPa and 160 GPa at fixed temperatures. At each temperature the resistance changes from zero to a finite value as $\mu_0 H$ increases due to the field-induced suppression of the superconductivity. $H_{c2}$ is defined as the intersection between an extrapolation of the resistance in the normal state and a line having the slope of the resistive transition at its middle point[28]. The same criterion is used across DC and pulsed field traces. The field value at the intersect with $R = 0$ axis, $H^*$, marks the onset of dissipation associated with melting of the vortex lattice. In contrast to the unconventional high-temperature superconductors (HTS), vortex

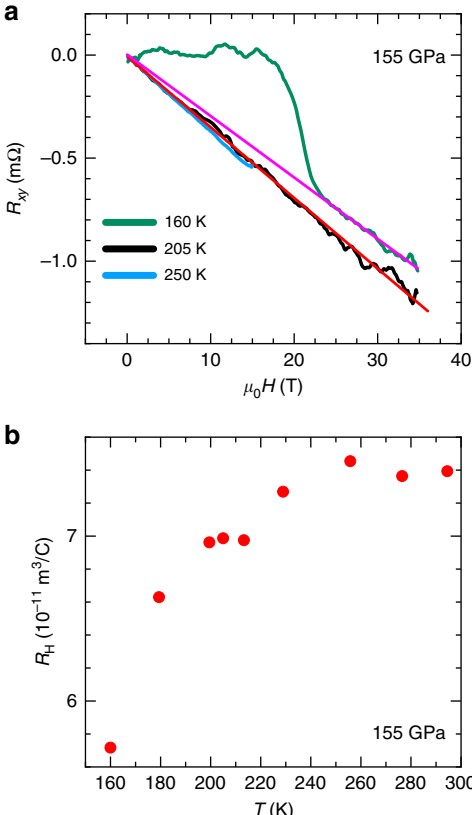

**Fig. 2** Hall-effect as a function of the field and for several temperatures. **a** Hall resistance $R_{xy}$ as a function of the magnetic field $\mu_0 H$ at a temperatures $T = 160$, 205, and 250 K for 155 GPa sample. For the trace collected at 160 K, the sample remains superconducting below 20 T, but recovers its linear-in-field behavior in the normal state above $H_{c2}$. Magenta and red lines are linear fits to the 160 K and 205 K curves, respectively. **b** Normal state Hall coefficient $R_H$ as a function of $T$ for the $H_3S$ sample pressurized under 155 GPa. Source data are provided as a Source Data file

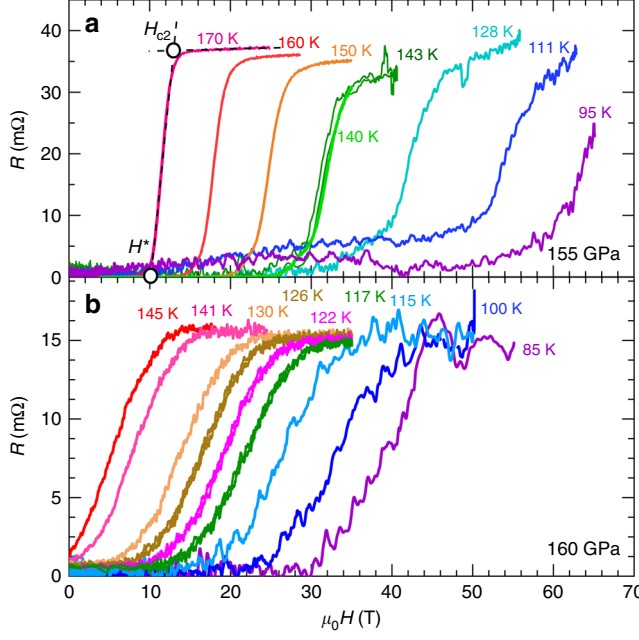

**Fig. 3** Upper critical fields for $H_3S$ as a function of the temperature. Resistance as a function of the field $\mu_0 H$ for the $H_3S$ samples under **a** $p = 155$ GPa and **b** 160 GPa and for several temperatures. This data was collected under continuous and pulsed fields. Two black dashed lines extrapolate the resistance in the normal state and the slope at the middle point of the resistive transition. The open circle symbol at the intersection of the two black dashed lines illustrates the position of $H_{c2}$. The open circle symbol at the intersect of the x-axis and the black dashed line illustrates the position of $H^*$ which marks the onset of dissipation. Source data are provided as a Source Data file

melting field in H₃S closely follows the upper critical filed, particularly in the more homogeneous 155 GPa sample (Supplementary Fig. 3).

The resistive onset has been found to match the thermodynamic $H_{c2}$ obtained through other experimental probes[29–31]. The resistive transition shifts to higher fields as the temperature is lowered while broadening slightly. The $H_{c2}(T)$ values obtained from pulsed field measurements agree with the curvature and the values extracted under DC fields. At very high pulsed magnetic fields the self-heating of the metallic DACs by the magnetic-field induced currents becomes noticeable at temperatures below 55 K.

Figure 4 shows the extracted $H_{c2}$ as a function of $T$ for both samples under $p = 155$ GPa and 160 GPa as a function of the reduced temperature $t = T/T_c$. To avoid the influence of possible superconducting vortex phases, we plot just the onset of their resistive transition. These phase boundaries were obtained from both isothermal field scans and from temperature scans under fixed magnetic fields. For both samples $H_{c2}(T)$ increases almost linearly upon decreasing the temperature under fields all the way up to 60 T yielding the slopes $|dH_{c2}/dT|_{T_c} = 0.62$ and 0.50 T K$^{-1}$ for the samples under 155 GPa and 160 GPa, respectively.

## Discussion

HTS hydrides are considered to be conventional BCS-type superconductors. At the same time, the linear $H_{c2}(T)$ dependence reported here is quite unusual for a weakly coupled BCS superconductor. The linear dependence of $H_{c2}(T)$ over an extended range of temperatures was also observed in the two-band superconductor MgB₂[32,33] and in the multigap Fe-pnictide superconductors for

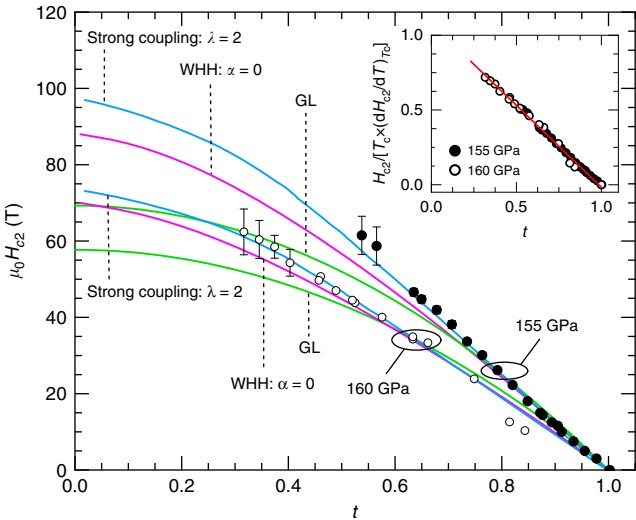

**Fig. 4** Superconducting phase diagram of H₃S. Upper critical fields $H_{c2}$ as a function of the reduced temperature $t = T/T_c$ for the H₃S samples under pressures $p = 155$ GPa (filled circles) and 160 GPa (open circles). Solid lines are fits to theoretical models: Ginzburg-Landau (GL), Werthamer, Helfand and Hohenberg (WHH), and a strong-coupling model using $λ = 2$ for the coupling strength parameter (reproduced from a prediction based on strong-coupling theory[40,41]). Although the GL formalism is mainly valid near $T_c$, we extrapolated the corresponding fit to $T = 0$ K to allow a visual comparison between all models. Error bars are standard deviation in the linear extrapolation of the resistance in the normal state and the slope of the resistive transition used to determine $H_{c2}$. Inset: upper critical fields for H₃S and for two different pressures plotted in reduced variables. The vertical axis is re-scaled by the product of $T_c$ and the slope of $H_{c2}(T)$ at $T_c$ as derived from the WHH fits for each sample. Red line is a linear fit. Source data are provided as a Source Data file

fields along certain, but not for all crystallographic orientations, and claimed to result from the orbital limiting effect[34–36]. Multigap effects might explain our phase diagram for H₃S in analogy with those systems. However, this multiband scenario contrasts with the linear in field Hall data in Fig. 2a, which indicates conduction dominated by a single type of carrier. Measurements of, for example, the penetration depth as a function of $T$ should help clarify the nature of the superconducting state in this system. The estimated weak coupling Pauli limiting fields for the samples under 155 GPa and 160 GPa are $H_p(0) \simeq 374$ T and 324 T, respectively. In the dirty limit, the $H_{c2}(0)$ values derived from the slope of the $H-T$ phase boundary at $T_c$ are, $H_{c2}^{orb}(0) = 0.69 dH_{c2}/dT|_{T_c} \times T_c = 86$ T for the sample under 155 GPa and 60 T for the one under 160 GPa. These latter values are much closer to the extrapolation of our experimental $H_{c2}(T)$ towards zero-temperature, when compared to the Pauli limiting values which differ by a factor of ~5, thus indicating that H₃S is an orbital-limited superconductor.

Subsequently, we fit our experimentally determined phase boundary to two different expressions in order to estimate the value of $H_{c2}$ at $T = 0$ K (see Methods). At temperatures near $T_c$, our experimental $H_{c2}$ values seem to follow the GL-expression, which is used to estimate $H_{c2}(0)$ in hydrides from the low field data[1,2], but at low temperatures they deviate from it considerably. This is particularly apparent for the sample under 160 GPa, for which we were able to reach smaller $t$ values, given that the GL theory is formulated for $(1 - t) \ll 1$. Fitting the upper critical fields to the WHH expression yields very small values for $α$ which points again to rather small orbital limiting fields relative to the Pauli limiting ones, with no apparent saturation in $H_{c2}(t)$ at low $t$s. We obtained the best fits to the WHH formula for $α$ values ranging between 0.0 and 0.3 for the sample under 155 GPa and $α = 0–0.2$ for the sample under 160 GPa. For such small values of $α$, the WHH formula is almost insensitive to the strength of the spin–orbit interaction. We obtained the superconducting coherence length by extrapolating the WHH fit to zero-temperature. This extrapolation of the $H_{c2}(T)$s yields $H_{c2}(T = 0) = 88$ T and 70 T for the H₃S samples pressurized up to 155 GPa and 160 GPa, respectively. Thus, we find coherence lengths of $ξ = 1.84$ nm and 2.12 nm for the samples under 155 GPa and 160 GPa, respectively, which are smaller than the initial estimate $ξ \simeq 4$ nm from first-principles calculations[15]. As for the effect of pressure on the H₃S samples, our experimental upper critical fields do not reveal any change in the relative strength of both pair-breaking mechanisms between both pressures. As seen in the inset of Fig. 4, the superconducting phase diagrams for both samples fall into a single curve when plotted in reduced units. What is remarkable is the linearity of the phase boundary as a function of $t$ all the way down to $t \sim 0.3$. Again, this resembles behavior reported, for example, for the Fe based superconductors[36].

This deviation from the conventional WHH formalism could be explained by the fact that it was formulated for the weak limit of electron–phonon coupling or for a coupling constant $λ \ll 1$. Recent ab initio calculations report substantial enhancement of $λ$ in H₃S due to the proximity to a structural instability between the $Im\text{-}3m$ and the $R3m$ crystal symmetries[37]. As the phonon frequencies $ω_p$ soften near this instability, the Cooper pairing becomes weaker at elevated temperatures by the thermal phonons, while it remains robust in the $T \to 0$ limit, leading to a relative enhancement in $H_{c2}(T)$ at low temperatures. The calculated $λ \sim 2$[9,37] would correspond to a nearly linear $H_{c2}(T)$ for $t > 0.3$ within the framework of the strong coupling model[40,41], which was also found to be applicable to the unusually strong electron–phonon coupled case of Bi-III[38]. Coincidentally, the strong coupling model also predicts a very

high $T_c \sim 300$ K–400 K for a lattice containing light atoms like H[40]. The fact is that our experimental $H_{c2}(T)$ values display a much better agreement with the temperature dependence predicted by the strong coupling model[40] with an electron–phonon coupling constant $\lambda = 2$.

In summary, we have investigated the temperature dependence of the upper critical fields of $H_3S$ under magnetic fields up to 65 T. At lower fields the phase boundary separating normal and superconducting states is relatively well described by the Werthamer, Helfand, and Hohenberg formula. Pronounced deviations from the WHH formula are observed at lower temperatures due to the linearity of $H_{c2}(T)$. Overall, the phase boundary between superconducting and metallic states indicates that the orbital-effect suppresses superconductivity over the entire temperature range. The linearity of $H_{c2}(T)$ observed over an extended range of reduced temperatures suggests that this system might indeed be a multigap superconductor as predicted theoretically and in analogy with similar results for the Fe-based superconductors. Alternatively, enhanced electron–phonon coupling and softening of hydrogen vibrational modes in the vicinity of a structural instability could also explain the deviation from the WHH predictions. We extract values for the coherence length $\xi$ ranging between $\xi = 1.84$ nm and 2.12 nm. Above the transition temperature the Hall-effect remains linear up to high magnetic fields indicating that $H_3S$ is a very good metal and that at high temperatures carrier conduction is dominated by electrons.

## Methods

**Sample preparation**. Two samples of $H_3S$ were synthesized in-situ inside the DACs using two different techniques. The first sample was prepared from condensed liquid $H_2S$ via a disproportionation reaction as described in ref. [1], the final pressure inside the sample was as high as 160 GPa. We refer to this sample as the 160 GPa sample. The second sample was synthesized directly from elemental sulfur and hydrogen at high pressures. For this sample a small piece of elemental sulfur (purity of 99.98%) with a lateral dimension of about 20 μm and a thickness of ~3 μm–5 μm was placed in a DAC having diamonds with culets of ~60 μm–70 μm. Excess hydrogen was introduced in the DAC at a gas pressure of ~130 MPa–150 MPa. Subsequently, the DAC was pressurized up to 150 GPa and then heated up to 1000 K with a YAG laser at room temperature to initiate the chemical reaction. The pressure increased slightly up to 155 GPa after the synthesis. The vibrational properties of the pressurized initial reactants, and of the synthesized products, were probed using a triple grating Raman spectrometer equipped with a HeNe laser having a wavelength of 633 nm. Sputtered gold electrodes were thoroughly isolated from the metal gasket by a layer made from magnesium oxide, calcium fluoride, and an epoxy glue mixture. This layer also prevented hydrogen penetration into the rhenium gasket. The pressure was estimated using the Raman shift of stressed diamond[39] and the $H_2$ vibron's wavenumber, previously calibrated in a separate experiment[42]. The uncertainty in these pressure values is estimated to be ±10%. Both scales indicated a pressure of 155 GPa after the laser heating. Both DACs with $H_3S$ samples remain under pressure after initial synthesis throughout the duration of the study. The sample under 155 GPa experienced slight decrease in resistance and increase in $T_c$ between July and September 2018 (Fig. 1), but no further change was noted.

**Magnetotransport measurements**. The magnetoresistance and the Hall signal was recorded using a conventional 4-probe configuration. Under continuous fields up to $\mu_0 H = 35$ T the resistance of the sample is measured using a commercial low-frequency (~13 Hz AC resistance bridge). A high-frequency (345 kHz) lock-in amplifier technique is used under pulsed fields up $\mu_0 H = 65$ T as follows. An AC current of $5 \times 10^{-4}$ (RMS) is applied to one pair of sample leads through an isolating radio-frequency transformer. The voltage drop across the second pair of leads was amplified using instrumentation amplifier and recorded at 30 MSPS rate with a high speed digitizer, which is synchronized with the AC current source. The corresponding in-phase and quadrature components of the AC signal are extracted from the recorded waveform by a software lock-in.

**Theoretical models for the upper critical fields**. First, the $H_{c2}(T)$ were fit to a conventional Ginzburg–Landau (GL) expression:

$$H_{c2}(T) = \frac{\phi_0}{2\pi\bar{\xi}(0)^2}(1 - t^2), \qquad (1)$$

where $\phi_0$ is the quantum of flux and $\bar{\xi}(0)$ is an average Ginzburg–Landau coherence length at $T = 0$.

**Table 1 Sample properties and results of WHH fit**

| Pressure | $T_c$ | $H_{c2}(0)$ | $dH_{c2}/dT|_{T_c}$ | $\xi$ |
|---|---|---|---|---|
| 155 GPa | 197 K | 88 T | 0.62 T K$^{-1}$ | 1.84 nm |
| 160 GPa | 174 K | 70 T | 0.50 T K$^{-1}$ | 2.12 nm |

Subsequently to the WHH formalism where the temperature dependence of $H_{c2}(T)$ defined by orbital and spin-paramagnetic effects in the dirty limit is given by[24]:

$$\ln\left(\frac{1}{t}\right) = \sum_{\nu=-\infty}^{\infty}\left\{\frac{1}{|2\nu+1|} - \left[|2\nu+1| + \frac{\bar{h}}{t} + \frac{(\alpha\bar{h}/t)^2}{|2\nu+1| + (\bar{h}+\lambda_{so})/t}\right]^{-1}\right\} \qquad (2)$$

where, $\bar{h} = (4/\pi^2)[H_{c2}(T)/(-dH_{c2}/dT)_{T_c}]$, $\alpha$ is the Maki parameter, and $\lambda_{SO}$ is the spin–orbit constant. The results of the fits are summarized in Table 1.

## Data availability

The data that support the findings of this study are available from the corresponding authors upon reasonable request. The source data underlying Figs. 1 through 4 and Supplementary Figs. 1 through 3 are provided as a Source Data file.

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

## Acknowledgements

We acknowledge A.V. Gurevich, H.H. Wen, and T. Timusk for useful discussions. This work was performed at the National High Magnetic Field laboratory, which is supported by National Science Foundation Cooperative Agreement No. DMR-1644779, and the State of Florida. S.M. acknowledges support from the FSU Provost Postdoctoral Fellowship Program. L.B. is supported by DOE-BES through award DE-SC0002613. This study was supported by JSPS KAKENHI, Specially Promoted Research (26000006). M.I.E. is thankful to the Max Planck community for the invaluable support and U. Pöschl for the constant encouragement.

## Author contributions

F.F.B., L.B. and M.I.E. designed the research; V.S.M., A.P.D., D.K., M.E., K.S. and M.I.E. prepared the samples, S.M., D.S., J.B.B., F.F.B., L.B. and M.I.E. performed high field measurements; S.M., F.F.B. and L.B. analyzed the data, and wrote the paper. All authors helped in writing the paper.

## Additional information

**Competing interests:** The authors declare no competing interests.

