## [Peer Review File · Nature Communications]

Reviewers' comments:

Reviewer #1 (Remarks to the Author):

Since the publication of the paper by A.P. Drozdov and M.I. Erements (Ref. 1 in the present manuscript) with a claim of a superconducting critical temperature over 200 K, many articles have been published, both experimental and theoretical. In the present paper, the authors present a study of the superconducting properties of H₃S at very high pressure (up to 180 GPa) and high pulsed magnetic field (65 T). The main results of this study are that under the present thermodynamic conditions, H₃S is a type II "classical" superconductor, whose properties may be described in the framework of the BCS theory. Its behavior can be correctly described as a strongly coupled superconductor. The experimental part is clearly a very remarkable achievement, and the scientific result an important one opening the route to the superconductivity at ambient temperature. As such, it should clearly be published in Nature Comm.

Some minor points should be addressed before the paper is published

1. Why is the resistance presented and not the resistivity of the sample? It is mandatory to give this explanation, at least in the "Methods"
2. The experimental setup is absolutely not described. The diamond anvil cell is more or less described, and I think that it is sufficient, since it was presented in Ref. 1, but the setup for the measurement at 65 T is clearly missing. It should be described in the "Methods" part, with eventually a scheme.
3. Although not very different from the claim that the latest pulsed field achievement was 4 GPa (ref. 25), a fast journey in the Web of Sciences shows that several results were published at higher pressure several years before that: R. Valiente et al. High Pressure Res. 29, 748 (2009) present results up to 7.5 GPa, or M. Milliot et al., Phys. Rev. B81, 205211 (2010) at 5 GPa and 53 T.
4. Fig. 1 should be improved: I do not see the necessity to use symbols. Use only thin lines in such a way one can see the difference between the various curves. This holds also for Fig. 2a
5. Concerning Fig. 1, a precision should be given also in the "Methods": Does this figure present results from a single loading, measured at different times or are they four different loadings, or ...?

Reviewer #2 (Remarks to the Author):

In this interesting manuscript, the authors report on measurements of the upper critical field in superconducting hydrogen sulfide at pressures of about 1.5 megabar. They were able to extend these measurements to pulsed magnetic fields of up to 65 Tesla, which enabled them to cover a large section of the field-temperature phase diagram, reaching down to about one third of the critical temperature. The key findings are that (i) the upper critical field extrapolates to values of the order of 100 Tesla, which is far below the expected Pauli (paramagnetic) limiting field, pointing towards orbital limiting as the dominant pair breaking mechanism, (ii) the temperature dependence of the upper critical field is almost linear and thereby deviates significantly from the standard WHH form, (iii) the temperature dependence of the upper critical field follows quite closely the expectations from a strong-coupling numerical calculation (Ref. 30) for an electron-phonon constant of about 2, in line with ab initio calculations for H₃S. They were also able to record the temperature dependence of the Hall coefficient of H₃S.

The measurements presented in this paper represent an important experimental achievement that deserves to be published in Nature Communications. Only few groups are able to carry out four-point resistivity and Hall effect measurements in an anvil pressure cell at megabar pressures on a material

which only forms at these pressures. To do these measurements under pulsed field conditions pushes the challenge to an even higher level. Demonstrating such measurements opens the door for further work on the newly discovered superconducting superhydride LaH₁₀. The data obtained in this study is urgently needed to verify numerical ab initio calculations, which have proven so important in this field of research.

The experimental results can make an important contribution to our understanding of superconductivity in H₃S and, by extension, in other hydrogen-rich superconductors. The authors' analysis and interpretation of the data is plausible, and if the authors could consider the following points, I would be in favour of rapid publication with minimal further changes:

- 1) The first sentence of the introduction: 'ongoing scientific crusade' is a rather extreme choice of words.
- 2) Fourth sentence on page 3: 'classical work by Clogston' suggests that the calculation was classical (rather than quantum), but I don't think this is what the authors have in mind.
- 3) Figures: could the authors not give absolute units for the resistivity? It should be possible, if systematic errors arising from geometry factors are estimated and stated. Also, if they could give the fit coefficients for ρ_0 and A, this may turn out to be useful in comparing to theory at a later stage.
- 4) Line 6 from the bottom on page 6: 'the slope *of* the resistive transition'
- 5) Bottom of page 8, discussion: the extrapolated fields using the WHH form, according to Fig. 4, appear to be more like 88 T and 70 T (red lines), not 97 T and 73 T, as given in the text. These latter values would correspond to the blue lines in Fig. 4, which however come from fitting the Bulaevskii approach to the data.
- 6) Importantly, when ξ is extracted (top of page 9), this represents an excellent opportunity to compare it to expectations from the band structure. This really ought to be done: is $\xi \sim vF/T_c$ consistent with what we know from ab initio calculations?
- 7) Line 7 on page 9: the measurements are taken down to $t \sim 0.3$, as far as I can see from Fig. 4, not $t \sim 0.25$. It may be better not to overstate the case here (and again a few lines further on).
- 8) The Ginzburg-Landau form for $H_{c2}(T)$ can really only be expected to hold close to T_c ; is it appropriate, then, to plot this curve down to temperatures much less than T_c ?

Reviewers' comments:

=====
Reviewer #1 (Remarks to the Author):
=====

Since the publication of the paper by A.P. Drozdov and M.I Eremets (Ref. 1 in the present manuscript) with a claim of a superconducting critical temperature over 200 K, many articles have been published, both experimental and theoretical. In the present paper, the authors present a study of the superconducting properties of H3S at very high pressure (up to 180 GPa) and high pulsed magnetic field (65 T). The main results of this study are that under the present thermodynamic conditions, H3S is a type II classical superconductor, whose properties may be described in the framework of the BCS theory. Its behavior can be correctly described as a strongly coupled superconductor. The experimental part is clearly a very remarkable achievement, and the scientific result an important one opening the route to the superconductivity at ambient temperature. As such, it should clearly be published in Nature Comm.

Our answer: We thank reviewer #1 for her/his careful review of our manuscript, the very positive comments on our work, and the useful comments on how to improve our manuscript. In the following we provide an answer to each point raised by this reviewer.

Some minor points should be addressed before the paper is published

1. Why is the resistance presented and not the resistivity of the sample? It is mandatory to give this explanation, at least in the Methods

Answer: In the supplementary information file, for the sample under a pressure of 155 GPa we report the resistivity measured through the van der Pauw method. However, we do not know the thickness of the sample very precisely. We estimated the thickness to be $t = (2 \pm 1) \mu m$, and with that we obtain a room temperature resistivity of $26 \mu\Omega cm$; one order of magnitude larger than that of copper (Supplementary Fig. 2).

2. The experimental setup is absolutely not described. The diamond anvil cell is more or less described, and I think that it is sufficient, since it was presented in Ref. 1, but the setup for the measurement at 65 T is clearly missing. It should be described in the "Methods" part, with eventually a scheme.

Answer: We agree with the referee and therefore we have updated the "METHODS" section, adding a detailed description of the setup used for the magnetotransport measurements, particularly in what concerns measurements under pulsed fields.

3. Although not very different from the claim that the latest pulsed field achievement was 4 GPa (ref. 25), a fast journey in the Web of Sciences shows that several results were published at higher pressure several years before that: R. Valiente et al. High Pressure Res. 29, 748 (2009) present results up to 7.5 GPa, or M. Milliot et al., Phys. Rev. B81, 205211 (2010) at 5 GPa and 53 T.

Answer: We thank the referee for pointing these manuscripts to us. We were able to locate a report concerning measurements under hydrostatic pressures up to 10 GPa and under pulsed fields by M. Millot, J.-M. Broto, and J. Gonzalez, Phys. Rev. B 78, 155125 (2008). This result is now mentioned in our manuscript.

4. Fig. 1 should be improved: I do not see the necessity to use symbols. Use only thin lines in such a way one can see the difference between the various curves. This holds also for Fig. 2a

Answer: We have updated Fig. 1 and Fig. 2 a, following the suggestion by the this reviewer. In Fig. 1, we also have placed the legends next to the curves. Hopefully, both figures are now clearer to the reader.

5. Concerning Fig. 1, a precision should be given also in the Methods: Does this figure present results from a single loading, measured at different times or are they four different loadings, or...?

Answer: All three curves for the sample under 155 GPa in Fig. 1 result from one single load but measured in different occasions. The curve for the sample under 160 GPa corresponds to another DAC which was pressurized up to higher values. We added a sentence to this effect in methods and updated the caption of Fig. 1.

=====
Reviewer #2 (Remarks to the Author):
=====

In this interesting manuscript, the authors report on measurements of the upper critical field in superconducting hydrogen sulfide at pressures of about 1.5 megabar. They were able to extend these measurements to pulsed magnetic fields of up to 65 Tesla, which enabled them to cover a large section of the field-temperature phase diagram, reaching down to about one third of the critical temperature. The key findings are that (i) the upper critical field extrapolates to values of the order of 100 Tesla, which is far below the expected Pauli (paramagnetic) limiting field, pointing towards orbital limiting as the dominant pair breaking mechanism, (ii) the temperature dependence of the upper critical field is almost linear and thereby deviates significantly from the standard WHH form, (iii) the temperature dependence of the upper critical field follows quite closely the expectations from a strong-coupling numerical calculation (Ref. 30) for an electron-phonon constant of about 2, in line with ab initio calculations for H3S. They were also able to record the temperature dependence of the Hall coefficient of H3S.

The measurements presented in this paper represent an important experimental achievement that deserves to be published in Nature Communications. Only few groups are able to carry out four-point resistivity and Hall effect measurements in an anvil pressure cell at megabar pressures on a material which only forms at these pressures. To do these measurements under pulsed field conditions pushes the challenge to an even higher level. Demonstrating such measurements opens the door for further work on the newly discovered superconducting superhydride LaH10. The data obtained in this study is urgently needed to verify numerical ab initio calculations, which have proven so important in this field of research.

The experimental results can make an important contribution to our understanding of superconductivity in H3S and, by extension, in other hydrogen-rich superconductors. The authors' analysis and interpretation of the data is plausible, and if the authors could consider the following points, I would be in favour of rapid publication with minimal further changes:

Answer: We sincerely thank reviewer # 2 for the evaluation of our manuscript, the strong support to our work, on for her/his recommendations on how to improve the manuscript. In the following we provide an answer to each point raised by this reviewer.

1) The first sentence of the introduction: 'ongoing scientific crusade' is a rather extreme choice of words.

Answer: We agree with the reviewer and we have substituted the word "crusade" with "endeavor".

2) Fourth sentence on page 3: 'classical work by Clogston' suggests that the calculation was classical (rather than quantum), but I don't think this is what the authors have in mind.

Answer: We agree with this reviewer and hence we have deleted the word "classical".

3) Figures: could the authors not give absolute units for the resistivity? It should be possible, if systematic errors arising from geometry factors are estimated and stated. Also, if they could give the fit coefficients for

ρ_0 and A , this may turn out to be useful in comparing to theory at a later stage.

Answer: In the new version of this manuscript we provide an estimation of the sample thickness $t = (2 \pm 1) \mu\text{m}$ and consequently of the resistivity at room temperature, i.e. $26 \mu\Omega \text{cm}$. In addition, we also provide a values for $\rho_0 = 8.6 \mu\Omega \text{cm}$ and $A = 2.1 \times 10^{-4} \mu\Omega \text{cm K}^{-2}$ (Supplementary Fig. 2)

4) Line 6 from the bottom on page 6: 'the slope *of* the resistive transition'

Answer: We thank the reviewer for pointing out this correction.

5) Bottom of page 8, discussion: the extrapolated fields using the WHH form, according to Fig. 4, appear to be more like 88 T and 70 T (red lines), not 97 T and 73 T, as given in the text. These latter values would correspond to the blue lines in Fig. 4, which however come from fitting the Bulaevskii approach to the data.

Answer: We thank the referee for the very careful review of our manuscript. We do agree that the 97 T and 73 T are from the extrapolation of the Bulaevskii approach to zero temperature. We have carefully extrapolated the WHH fit to zero temperature and obtained the values of 88 T and 70 T for the samples under 155 GPa and 160 GPa. We have corrected the manuscript accordingly.

6) Importantly, when ξ is extracted (top of page 9), this represents an excellent opportunity to compare it to expectations from the band structure. This really ought to be done: is $\xi \sim v_F/T_c$ consistent with what we know from ab initio calculations?

Answer: We thank the referee for reminding us of this excellent point. As the referee pointed out, in the clean limit $\xi = \frac{\hbar v_F}{\pi \Delta(0)}$. From the first principle calculations provided in reference Phys. Rev. B **91**, 060511(R) (2015), the Fermi velocity for H_3S is estimated to be $0.25 \times 10^8 \text{ cm/s}$ and $\Delta(0) \sim 40 \text{ meV}$, which yields a coherence length of $\sim 40 \text{ \AA}$. This value is almost twice the one we got from our experimental measurements. We added one sentence into the manuscript to compare the coherence length suggested by band structure calculations with the value of ξ resulting from our experimental measurements of the upper critical field as a function of the temperature.

7) Line 7 on page 9: the measurements are taken down to $t \sim 0.3$, as far as I can see from Fig. 4, not $t \sim 0.25$. It may be better not to overstate the case here (and again a few lines further on).

Answer: We agree with the referee and we have corrected the minimum experimental value of t in both places.

8) The Ginzburg-Landau form for $H_{c2}(T)$ can really only be expected to hold close to T_c ; is it appropriate, then, to plot this curve down to temperatures much less than T_c ?

Answer: The Ginzburg-Landau extrapolation was added to facilitate the comparison with previous publications, where the GL expression is used to estimate $H_{c2}(0)$ in hydrides. We have added a sentence in the caption of Fig. 4 and also in the main text to clarify this point.

=====
Reviewer #3 (Remarks to the Author):

=====
Fig. 1; the present legend is difficult to understand. Try to make the figure more reader friendly.

Answer: We updated Fig. 1, according to the suggestions from Reviewer #1, and also placed the legends next to the each curve. We hope that with these changes Fig. 1 became more reader friendly.

Fig. 2a; the position of a is difficult to catch. $R_{xy} = 0$ up to 15 T for the 160 K data is superconducting? If yes, this should be explained somewhere in the text. It is recommended to put $P = 155 \text{ GPa}$ in the figure.

Answer: Yes, for the trace collected at 160 K, the sample is superconducting below 20 T and thus the Hall resistance $R_{xy} = 0$. We added a sentence into the caption of Fig. 2 and a guide line for this trace to make this point clearer. We also added “155 GPa” within the figure to clarify the value of the pressure applied to this particular sample.

Fig 3; What is the experimental definition of H_{c2} ?

Answer: In the main text we indicate that H_{c2} corresponds to the onset of the resistive transition. It is obtained by extrapolating the resistance in the normal state until it intersects a line that extrapolates the slope at the middle point of the resistive transition. To help clarify the definition of H_{c2} , we have added into Fig. 3 two black dashed lines and two markers (\circ) indicating the position/definition of the critical field at the intersection of the two black dashed lines as H_{c2} , and the slope with the x-axis as H^* , marking the onset of dissipation. We also added a Supplementary Figure 3, which summarizes H_{c2} and H^* temperature dependencies.

Fig. 4: What is the final value of H_{c2} used for the inset? Is it possible to add a table showing P , T_c , and $H_{c2}(0)$? This makes readers understand the result immediately, even without reading the text.

Answer: For the inset in Fig. 4, we used the H_{c2} values obtained from the WHH formalism. We also corrected the label of the vertical axis on the inset and in the caption - the axis is scaled by the normalized gradient of $H_{c2}(T)$ at T_c rather than by $H_{c2}(T)$ at $T = 0$. We thank this reviewer for her/his second suggestion. We have now added a table in the “METHODS” section, summarizing the results of the WHH fit. We also added a Supplementary Figure 1, explaining the choice of T_c for $H_{c2}(T)$ fit for the 155 GPa sample